European population trends and current conservation status of an endangered steppe-bird species: the Dupont’s lark Chersophilus duponti

Gómez-Catasús Julia 1 julia.gomez@uam.es
Pérez-Granados Cristian 1 2
Barrero Adrián 1
Bota Gerard 3
http://orcid.org/0000-0001-9712-1957 Giralt David 3
http://orcid.org/0000-0003-3045-5498 López-Iborra Germán M. 2
http://orcid.org/0000-0001-6205-386X Serrano David 4
http://orcid.org/0000-0001-6326-8942 Traba Juan 1
1 Terrestrial Ecology Group (TEG-UAM), Department of Ecology, Universidad Autónoma de Madrid , Madrid , Spain
2 Multidisciplinary Institute for Environmental Studies “Ramón Margalef”, Department of Ecology, Universidad de Alicante , Alicante , Spain
3 Biodiversity and Animal Conservation Lab, Forest Sciences Center of Catalonia (CTFC) , Solsona, Catalonia , Spain
4 Department of Conservation Biology, Estación Biológica de Doñana (EBD-CSIC) , Sevilla , Spain
Pimm Stuart
Electronic publication date: 2018 Sep 19
Publication date: 2018
Volume: 6
Electronic Location ID: e5627
Received 2018 May 4; Accepted 2018 Aug 23
Copyright: © 2018 Gómez-Catasús et al.
Copyright year: 2018
Copyright holder: Gómez-Catasús et al.
License: This is an open access article distributed under the terms of the Creative Commons Attribution License, which permits unrestricted use, distribution, reproduction and adaptation in any medium and for any purpose provided that it is properly attributed. For attribution, the original author(s), title, publication source (PeerJ) and either DOI or URL of the article must be cited.
License URL: https://creativecommons.org/licenses/by/4.0/

Keywords: Listing criteria, Shrub-steppes, Trend analysis, Threat categories

Funding: The Education, Youth and Sport Bureau (Madrid Regional Government) and the European Social Fund for the Youth Employment Initiative (reference number PEJ15/AMB/AI-0059) The European Social Fund for the Youth Employment Initiative (reference number PEJ15/AMB/AI-0059) Excellence Network Remedinal 3CM (S2013/MAE2719) Madrid Regional Government; the project ‘Scientific basis for the National Conservation for Dupont’s Lark in Spain’ Biodiversity Foundation, of the Ministry of Agriculture and Fisheries, Food and Environment; the Life Ricotí project (LIFE15-NAT-ES-000802) European Commission; and the BBVA-Ricotí project, granted by the BBVA Foundation This work was supported by the Education, Youth and Sport Bureau (Madrid Regional Government) and the European Social Fund for the Youth Employment Initiative (reference number PEJ15/AMB/AI-0059). This research is a contribution to the Excellence Network Remedinal 3CM (S2013/MAE2719), supported by Madrid Regional Government; the project ‘Scientific basis for the National Conservation for Dupont’s Lark in Spain’, funded by the Biodiversity Foundation of the Ministry of Agriculture and Fisheries, Food and Environment; the Life Ricotí project (LIFE15-NAT-ES-000802), supported by the European Commission; and the BBVA-Ricotí project, funded by the BBVA Foundation. There was no additional external funding received for this study. The funders had no role in study design, data collection and analysis, decision to publish, or preparation of the manuscript.

==============================
Background

Steppe-birds face drastic population declines throughout Europe. The Dupont’s lark Chersophilus duponti is an endangered steppe-bird species whose European distribution is restricted to Spain. This scarce passerine bird could be considered an ‘umbrella species’, since its population trends may reveal the conservation status of shrub-steppes. However, trends for the Spanish, and thus European, population of Dupont’s lark are unknown. In this work, we evaluated Dupont’s lark population trends in Europe employing the most recent and largest compiled database to date (92 populations over 12 years). In addition, we assessed the species threat category according to current applicable criteria (approved in March 2017) in the Spanish catalogue of threatened species (SCTS), which have never been applied to the Dupont’s lark nor to any other Spanish species. Finally, we compared the resulting threat categories with the current conservation status at European, national and regional levels.

Methods

We fitted switching linear trend models (software TRIM—Trends and Indices for Monitoring data) to evaluate population trends at national and regional scale (i.e. per Autonomous Community) during the period 2004–2015. In addition, the average finite annual rate of change (λ¯) obtained from the TRIM analysis was employed to estimate the percentage of population size change in a 10-year period. A threat category was assigned following A1 and A2 criteria applicable in the SCTS.

Results

Trends showed an overall 3.9% annual decline rate for the Spanish population (moderate decline, following TRIM). Regional analyses showed high inter-regional variability. We forecasted a 32.8% average decline over the next 10 years. According to these results, the species should be listed as ‘Vulnerable’ at a national scale (SCTS). At the regional level, the conservation status of the species is of particular concern in Andalusia and Castile-Leon, where the species qualifies for listing as ‘Endangered’.

Discussion

Our results highlight the concerning conservation status of the European Dupont’s lark population, undergoing a 3.9% annual decline rate. Under this scenario, the implementation of a wide-ranging conservation plan is urgently needed and is vital to ensuring the conservation of this steppe-bird species. The role of administrations in matters of nature protection and the cataloguing of endangered species is crucial to reverse declining population trends of this and other endangered taxa.

Introduction

Steppes and pseudo-steppes are two of the most important habitats for the preservation of bird diversity, since 55% of European bird species listed on the IUCN Red List are highly dependent on these habitats (Burfield, 2005). Moreover, 83% of steppe-bird species show an unfavourable conservation status in Europe (Burfield & Van Bommel, 2004; Burfield, 2005). This is a consequence of the accelerated process of land use changes occurring in steppe-like habitats, with dramatic consequences for steppe-bird populations across Europe (Benton, Vickery & Wilson, 2003; Burfield & Van Bommel, 2004; Santos & Suárez, 2005). The main habitat-related threats, and therefore drivers of steppe-bird population declines are: (i) changes in land use (afforestation, new crops, infrastructure development, mining, rubbish dumps; Burfield, 2005; Laiolo & Tella, 2006a; Gómez-Catasús et al., 2016; Gómez-Catasús, Garza & Traba, 2018); (ii) agricultural intensification (landscape homogenization, irrigation, increase in the use of agrochemicals; Donald, Green & Heath, 2001; Benton, Vickery & Wilson, 2003; Brotons, Mañosa & Estrada, 2004; Burfield, 2005); and (iii) land abandonment and changes in agriculture and livestock management (Madroño, González & Atienza, 2004; Burfield, 2005).

Spain is the stronghold for steppe-birds in Western Europe, harbouring a large proportion of their total European breeding population (Burfield, 2005). However, most of the Spanish steppe-bird populations declined during the 1990–2000 period (Burfield, 2005) and later (BirdLife International, 2015). A species of particular conservation concern is the Dupont’s lark Chersophilus duponti (Vieillot, 1820), identified amongst the 65 priority bird species inhabiting steppes (Burfield & Van Bommel, 2004) and one of the scarcest passerine birds with a rather restricted distribution range in Europe. The species is classified as ‘Near Threatened’ on the IUCN Red List (BirdLife International, 2017) and as ‘Vulnerable’ on both the European Red List of Birds (BirdLife International, 2015) and on the Spanish catalogue of threatened species (SCTS; Royal Decree 139/2011, 4th February). Its European geographic range is restricted to Spain spreading over 1,480 km2 (Suárez, 2010), and its population has been estimated at 1,300–2,400 breeding pairs (Garza, Traba & Suárez, 2003; Tella et al., 2005; Suárez, 2010). The European population of Dupont’s lark qualifies for consideration as an Evolutionary Significant Unit (sensu Moritz, 1994; Casacci, Barbero & Balletto, 2014), as they are isolated and genetically and morphologically differentiated from the African populations (García et al., 2008; Suárez, 2010).

The species inhabits flat (<10–15% of slope) shrub-steppes, avoiding dry pastures and cereal fields (Garza et al., 2005; Seoane et al., 2006; Pérez-Granados, Lopez-Iborra & Seoane, 2017). Habitat fragmentation and land use changes, common issues in steppe ecosystems, have been documented as the main threats to the species (Tella et al., 2005; Íñigo et al., 2008; Garza & Traba, 2016; Pérez-Granados, Osiejuk & López-Iborra, 2016; Gómez-Catasús, Garza & Traba, 2018).

European Dupont’s lark population trends have been previously assessed globally (Suárez, 2010) or in a sample of populations (Tella et al., 2005; Pérez-Granados & López-Iborra, 2013, 2014). Despite the fact that the results of all of these studies showed declining population trends, none of them derived population change estimates using appropriate statistical methods. Moreover, current trends for the whole Spanish (and European) population are unknown, so an updated and rigorous assessment is needed. This updated information would allow an assessment of the conservation status of the species based on a formal set of criteria at two spatial scales: national and regional (i.e. per Autonomous Community where the species is present). The importance of both spatial scales relies on the jurisdiction of the Spanish Autonomous Communities in nature protection and, specifically, in listing and cataloguing endangered species (Law 42/2007, 13th December). The Spanish Ministry of Agriculture and Fisheries, Food and Environment has the jurisdiction to list the species at a national scale in the SCTS (Law 42/2007, 13th December) and to elaborate the National Conservation Strategy of endangered species. On the other hand, each Autonomous Community is legally bound to list species in its regional catalogue of threatened species (RCTS), at least with the same category as at the national level. In addition to this, they have the competence to elaborate and implement both conservation and recovery plans for those species classified as ‘Vulnerable’ and ‘Endangered’, respectively. Thus, regional population trends are crucial to assess whether species conservation status is of particular concern in specific regions and if the category of threat should be increased in the pertinent catalogues.

The species included in the SCTS were listed in 2011 (Royal Decree 139/2011), but listing criteria applicable in the SCTS were modified in March 2017 (Royal Decree 139/2011, 4th February; Resolution 6th March 2017), to accommodate those of the IUCN (2012). However, the conservation status of catalogued species in the SCTS has not been reviewed since this modification. To our knowledge, new criteria have never been applied to the Dupont’s lark nor to any other Spanish species and, therefore, an assessment of the category of threat assigned under the new criteria is needed.

In this work, we aimed to evaluate Dupont’s lark population trends during the 2004–2015 period at both national and regional scales, using the largest database ever compiled. We also carried out a comprehensive assessment of the conservation status of the Dupont’s lark according to quantitative threshold criteria of reduction in population size (A1 and A2 criteria, see below) under the SCTS (Resolution 6th March 2017). Finally, we aimed to assess whether the current threat category of the species at European (European Red List of Birds), national (SCTS) and regional levels (RCTS) agrees with Dupont’s lark populations trends.

Materials and Methods

Data collection

The ethics committee of Animal Experimentation of the Autonomous University of Madrid as an Organ Enabled by the Community of Madrid (Resolution 24th September 2013) for the evaluation of projects based on the provisions of Royal Decree 53/2013, 1st February, has provided full approval for this purely observational research (CEI 80-1468-A229).

We compiled data for 92 Dupont’s lark populations during the 2004–2015 period. This dataset comprised 41.6% of the known Spanish population (221 populations surveyed during the II National Survey 2004–2006; Suárez, 2010) and included all of the Autonomous communities where the species occurs (Fig. 1) (Suárez, 2010). The time series addresses a temporal range between one and 12 years (mean ± SD = 5.36 ± 2.77 years). We considered a single population to be all individuals living in patches with potential habitat for the species (i.e. short shrub with slopes lower than 15%; Garza et al., 2005) separated by less than one km.

Figure 1 Dupont’s lark distribution in Spain according to Suárez, 2010 (light grey) and Dupont’s lark populations included in this study (black).

The names of the Autonomous Communities where the species is present, are shown. The arrow refers to an isolated region belonging to the Community of Valencia. AN, Andalusia; AR, Aragon; CA, Catalonia; CL, Castile-Leon; CM, Castile-La Mancha; CV, Community of Valencia; NA, Navarre; RM, Region of Murcia.

The Dupont’s lark population size is difficult to quantify due to the extremely shy and elusive behaviour of the species and the concentration of singing activity mainly before dawn. Therefore, surveys of the species rely on auditory contacts. Bird censuses were carried out during the breeding season (March–June depending on phenological differences; Garza, Suárez & Carriles, 2010) approximately 1 h before dawn, when singing activity peaks, and they spanned around 1 h. Birds were counted by linear transects (500 m inner belt width; Garza, Suárez & Carriles, 2010) or by territory mapping (Bibby et al., 2000), with the two methods producing similar population size estimates (Pérez-Granados & López-Iborra, 2017). A slightly different census method, consisting of a network of point counts, was performed in Catalonia (CA) and Region of Murcia (RM) monitored populations (comprising less than 5% of all populations). The counting method remained constant throughout the study period within each region, making interannual data comparable. Linear transects were designed to cover the whole population (Suárez, 2010), and were walked at a constant speed, georeferencing singing males with a GPS and noting all males singing simultaneously. Transects were walked once per year under the linear transect method and two to four times per year under the mapping method. In the case of the territory mapping method, number of territories per population was estimated by mapping all records and gathering accumulated observations from different surveys, taking into account birds heard simultaneously (Garza, Suárez & Carriles, 2010; Pérez-Granados & López-Iborra, 2017). Population size estimates refer to the minimum number of territories (mapping method), or minimum number of recorded males (line transect method and point counts) per population. Lastly, we considered a population as extinct when the species was not detected in at least the last two surveys (hereafter local extinction event).

Trend analysis

Changes in population estimates were evaluated using the software TRIM (Trends and Indices for Monitoring data. TRIM v. 3.54. Pannekoek & Van Strien, 2006a). TRIM fits log-linear models and was employed because: (i) it allows the analysis of time series with the absence of data for some years, a common issue in long-term time series; and (ii) it takes into account overdispersion and serial correlation of data (Pannekoek & Van Strien, 2005). TRIM calculates indices that represent the effect of change between years, which indicates relative variation of the total population size. Two types of indices are estimated: (i) model-based indices, which are the values predicted by the model; and (ii) imputed indices, which equal the observed count if an observation is made and the model prediction for missing counts (Pannekoek & Van Strien, 2005). Dissimilarity between the two indices reflects a mismatch between observed (i.e. imputed indices) and model predictions (i.e. model-based indices) and, therefore, a lack of fit of the statistical model applied. Imputed indices are employed to estimate a mean annual change rate since they show a more realistic course in time (Pannekoek & Van Strien, 2006b) and a trend category is assigned (Pannekoek & Van Strien, 2006a). This technique has been broadly employed for the analysis of temporal series in bird populations (Paradis et al., 2002; Wretenberg et al., 2007; Delgado et al., 2009; Gómez-Catasús, Garza & Traba, 2018).

We fitted switching linear trend models to evaluate both national and regional Dupont’s lark trends during the period 2004–2015. TRIM employs a stepwise selection of change-points in trends using Wald-tests for the significance of change-points. When the difference between parameters before and after a change-point does not differ from zero (default significance threshold: 0.2), the corresponding change-point is removed from the model complying with the parsimony principle (Pannekoek & Van Strien, 2005). The best-fit models were selected according to Goodness-of-fit tests (Likelihood ratio (LR) test and Chi-squared) and Akaike information criterion (AIC). A model with a significance value greater than 0.05 indicates that the data fit a Poisson distribution and, therefore, the model can be accepted. Indices, overall slope and Wald tests remain reliable in case of lack-of-fit (Pannekoek & Van Strien, 2005). In case of overdispersion or serial correlation (default TRIM threshold: >3.0 and >0.4, respectively; Pannekoek & Van Strien, 2006b), the Wald-test for the significance of slope was employed (Pannekoek & Van Strien, 2005). While the whole set of 92 populations was used to analyse national trends, regional subsets were subsequently extracted to analyse regional trends (see Table 1 for sample size in each region).

Table 1 Results of regional switching linear trend models through the time series 2004–2015.

	AN	AR	CA	CL	CM	CV	NA	RM	
Number of populations	12	10	1	29	26	8	3	3	
Local extinction events	6	0	0	5	5	3	1	0	
Missing values (%)	38.2	81.6	58.3	49.1	63.1	44.8	63.9	47.2	
Annual change rate (%)	−10.9	+1.5	−8.7	−8.4	+1.5	−2.5	−1.1	+2.6	
95% confidence interval	[−16.2; −5.7]	[−2.3; +5.2]	[−35.5; +18.2]	[−10.0; −6.7]	[−2.1; +5.1]	[−5.7; +0.7]	[−7.9; +5.6]	[−2.2; +7.5]	
TRIM trenda	Steep decline	Uncertain	Uncertain	Steep decline	Uncertain	Uncertain	Uncertain	Uncertain	
Wald-test change rate	–	–	0.04	–	–	–	–	–	
p-value	–	–	>0.05	–	–	–	–	–	
Goodness-of-fit test	
Chi-squared (χ2)	98.98	11.56	–	187.13	152.34	63.00	2.00	4.98	
p-value χ2	0.0158*	>0.05	–	<0.01	<0.01	0.0152*	>0.05	>0.05	
Likelihood ratio (LR)	100.81	11.85	–	211.67	139.36	63.53	2.24	5.44	
p-value LR	0.0115*	>0.05	–	<0.01	<0.01	0.0136*	>0.05	>0.05	
AIC	–41.19	–10.15	–	–74.33	–24.64	–18.47	–3.76	–18.56	
Overdispersion	1.39	1.01	6.67	1.29	1.69	1.43	0.98	0.23	
Serial correlation	0.09	−0.18	−0.06	0.39	0.20	0.30	−	0.06	
Notes:

AN, Andalusia; AR, Aragon; CA, Catalonia; CL, Castile-Leon; CM, Castile-La Mancha; CV, Community of Valencia; NA, Navarre; RM, Region of Murcia.

p-values of accepted models are marked in bold.

p-values of models near to acceptance threshold are marked with asterisk (*).

a Trend classification attending to TRIM criteria (Pannekoek & Van Strien, 2006b).

Threat category

We evaluated the Dupont’s lark category of threat according to A1 (population size reduction over the last 10 years or three generations, whichever is longer) and A2 (population size reduction within the next 10 years or three generations, whichever is longer) criteria applicable in the SCTS. We employed a 10-year period because it is longer than three generations (generation length of the Dupont’s lark is estimated at 2.5 years; Íñigo et al., 2008). We used recent trends to forecast future population trends of the species, since its geographic range reduction (Traba et al., 2016) and the lack of conservation measures (Tella et al., 2005; Suárez, 2010; Pérez-Granados & López-Iborra, 2014) predict similar population trends in the following years.

The average finite annual rate of change (λ¯) during the study period was obtained from the TRIM analysis. This is a multiplicative factor representing the average growth rate over one time-step (i.e. 1 year). When this multiplicative factor is λ¯<1 the population decreases; when λ¯=1 the population remains stable; and when λ¯>1 the population increases. The λ¯ value was employed to estimate the percentage of population size change in a 10-year period following the equation below: Percentage of change in a 10-year period (%) =(λ¯10−1 )×100

We assigned a threat category according to population size reduction estimated over the last 10 years (A1 criterion; ‘Endangered’ ≥70% ‘Vulnerable’ ≥50%) and forecasted in the next 10 years (A2 criterion; ‘Endangered’ ≥50% ‘Vulnerable’ ≥30%) at both national and regional scales. Lastly, categories were compared with the current threat categories for the Dupont’s lark on the European Red List of Birds, the SCTS and the RCTS.

Results

Spanish (European) population trend

The best switching linear trend model for all Dupont’s lark populations did not fit a log-linear distribution (Chi-square, χ2 = 684.92, df = 389, p < 0.001; LR = 722.30, df = 389, p < 0.001; AIC = −55.70). Overdispersion and serial correlation values were relatively low (1.70 and 0.34, respectively), but 55.8% of counts were missing values. The stepwise procedure revealed six significant change-points in trends (Fig. 2 and Table S1). The population size index experienced an overall 41.4% decline (95% confidence intervals (CI) [−50.5 to −32.4]) from 2004 to 2015. Furthermore, the extinction of 20 populations, which represents 21.7% of the set of study populations, was registered in this period (Table S2). The overall slope parameter showed a 3.9% annual decrease (95% CI [−4.9 to −2.8]), which corresponds to a moderate decline according to TRIM criteria (Pannekoek & Van Strien, 2006a).

Figure 2 Imputed (grey continuous line) and predicted (black continuous line) population size indices estimated by the switching linear trend model for 92 Dupont’s lark populations during the 2004–2015 period.

Time-points incorporated in the model as significant change-points on population trends are marked with asterisk (*). The 95% confidence intervals (striped grey lines) are depicted.

Regional population trends

Regional trends showed high variability between regions (Table 1; Fig. 3). Switching linear trend models for Aragon (AR), Navarre (NA) and RM populations fitted a log-linear distribution (χ2 and LR p-values > 0.05), while Goodness-of-fit tests for models of Andalusia (AN) and Community of Valencia (CV) were near acceptance values (χ2 and LR p-values > 0.01; Table 1). However, Castile-La Mancha (CM) and Castile-Leon (CL) models did not fit a log-linear distribution (χ2 and LR p-values < 0.01; Table 1). Overdispersion and serial correlation values were of less concern for all models except for CA (Table 1), so we relied on Wald-tests for best-model selection. The proportion of missing values was higher than 50% for AR, CM, CA and NA models, and sample sizes were small for all regions (i.e. less than 15 populations) except for CM and CL (Table 1). Significant change-points in slope were incorporated in all models except for AR, CA and NA (Fig. 3 and Table S3), due to a constant slope in trends throughout the study period (AR and NA) or to sparse data that hindered the fitting of a switching linear trend model (CA). Trend analyses showed mean overall decreases in AN (66.8%), CM (59.0%), CL (51.1%), CA (42.9%), CV (30.1%) and NA (11.0%) during the 2004–2015 period (Table 2). However, mean overall trends were positive in AR (18.7%) and RM (55.2%) populations (Table 2). Average annual change rates showed a steep decline for AN and CL populations, greater than 5% per year (Table 1; Fig. 3). Population trends of AR, CA, CM, CV, NA and RM were classified as uncertain (Table 1). Local extinction events were registered mainly in AN (6), CL (5) and CM (5) (Table S2). The only known population in CA (Alfés) and one population in AN (Sierra de Gador-Llano de los Brincos) experienced a local extinction event followed by a recolonization event.

Figure 3 Imputed (grey continuous line) and predicted (black continuous line) population size indices estimated by switching linear trend models during the 2004–2015 period for each Autonomous Community.

(A) Andalusia; (B) Aragon; (C) Catalonia; (D) Castile-Leon; (E) Castile-La Mancha; (F) Community of Valencia; (G) Navarre; (H) Region of Murcia. Time-points incorporated in models as significant change-points on population trends are marked with asterisk (*). The 95% confidence intervals (striped grey lines) are depicted.

Table 2 Assessment of Dupont’s lark threat category.

	Overall change rate (%) from 2004 to 2015	Average annual change rate (%)	Current category of threat	Change rate for 10 years (%)	Category of threat—A1 criterion	Category of threat—A2 criterion	
AN	−66.8 [−85.4; −48.2]	−10.9 [−16.2; −5.7]	VUa	−68.5 [−82.9; −44.4]	VU [EN; None]	EN [EN; VU]	
AR	+18.7 [−29.7; +67.1]	+1.5 [−2.3; +5.2]	SHAb	+16.1 [−20.8; +66.0]	None [None; None]	VU* [VU*; VU*]	
CA	−42.9 [−178.0; +92.2]	−8.7 [−35.5; +18.2]	–	+42.4 [−98.3; +2.9 × 103]	None [EN; None]	VU* [EN; VU*]	
CL	−51.1 [−61.4; −40.8]	−8.4 [−10.0; −6.7]	–	−58.4 [−65.1; −50.0]	VU [VU; VU]	EN [EN; EN]	
CM	−59.0 [−78.9; −39.1]	+1.5 [−2.1; +5.1]	VUc	+16.1 [−19.1; +64.4]	None [None; None]	VU* [VU*; VU*]	
CV	−30.1 [−60.3; +0.1]	−2.5 [−5.7; +0.7]	VUd	−22.4 [−44.4; +7.2]	None [None; None]	VU* [VU; VU*]	
NA	−11.0 [−78.0; +56.0]	−1.1 [−7.9; +5.6]	SHAe	−10.5 [−56.1; +72.4]	None [VU; None]	VU* [EN; VU*]	
RM	+55.2 [−52.4; +162.8]	+2.6 [−2.2; +7.5]	VUf	+29.3 [−19.9; +106.1]	None [None; None]	VU* [VU*; VU*]	
Spain	−41.4 [−50.5; −32.4]	−3.9 [−4.9; −2.8]	VU	−32.8 [−39.5; −24.7]	None [None; None]	VU [VU; None]	
Notes:

Overall and average annual change rate obtained from trend analysis and current threat category at National and Regional Catalogues of Endangered Species, are shown. In addition, population size change in a 10-year period and corresponding threat category attending to A1 and A2 criteria applicable in the SCTS (Resolution 6th March 2017) are provided. The 95% confidence intervals are shown in brackets. Threat categories: sensitive to habitat alteration (SHA), vulnerable (VU) and endangered (EN). AN, Andalusia; AR, Aragon; CA, Catalonia; CL, Castile-Leon; CM, Castile-La Mancha; CV, Community of Valencia; NA, Navarre; RM, Region of Murcia.

a Decree 23/2012 of 14 February 2012.

b Decree 49/1995 of 28 March 1995.

c Decree 33/1998 of 5 May 1998.

d Decree 32/2004 of 27 February 2004.

e Decree 563/1995 of 27 November 1995.

f Law 7/1995 of 21 April 1995.

* Minimum category of threat in accordance to the category of threat in the SCTS (Law 42/2007, 13th December).

Threat category

According to the estimated mean annual rate of change (−3.9%), the Dupont’s lark population size in Spain has been reduced on average by 32.8% over the last 10 years and we expect it to be reduced by the same percentage in the next 10 years (Table 2). This reduction in population size does not entail the classification of the Dupont’s lark at any category of threat in Spain according to A1 criterion (Table 2). However, the Dupont’s lark should be classified as ‘Vulnerable’ on the SCTS according to A2 criterion (Table 2).

Regional analyses showed that the species should be classified as ‘Vulnerable’ in AN and CL according to past population trends (A1 criterion) while no category of threat is assigned in the rest of the Regional Catalogues (Table 2). Nevertheless, the species should be classified at least as ‘Vulnerable’ in all the Regional Catalogues according to forecasted population declines (A2 criterion) and Spanish legislation (Table 2). Specifically, the species should be upgraded to ‘Endangered’ in AN and CL in agreement with A2 criterion (Table 2).

Discussion

Our results provide evidence of concerning trends for the Spanish Dupont’s lark population, the remaining bastion of this endangered steppe-bird in Europe. The species exhibited an estimated annual decline rate of 3.9% and an overall 41.4% decline over 12 years (2004–2015). This result agrees with previously described trends for the Dupont’s lark (Tella et al., 2005; Pérez-Granados & López-Iborra, 2013) and for most of steppe-bird species in the Iberian Peninsula (Burfield, 2005; BirdLife International, 2015). Previous work on Spanish Dupont’s lark population trends suggested a 31.5% decline in 16 years (N = 34 populations; Tella et al., 2005) and a 70% decline in 12.5 years (N = 33 populations; Pérez-Granados & López-Iborra, 2014). In particular areas of its Spanish distribution, positive trends have been previously estimated in AR (N = 7) and RM (N = 2), whereas declining population trends have been described for AN (N = 4), CM (N = 6), CL (N = 6), CV (N = 6) and NA (N = 2) populations (a decline between 22% and 98% in 12.5 years; Pérez-Granados & López-Iborra, 2014). The novelty of the present work relies on the employment of a rigorous statistical method and on the incorporation of a greater number of Dupont’s lark populations (N = 92) covering a wider range of its European distribution.

In this study, we compiled the most updated database for Dupont’s lark population trends. We considered that our geographical coverage is representative of the Spanish (European) distribution, leading to reliable results for the population trend analysis. Most regions were significantly represented in this sample, ranging from 43% of the total regional population for CL, to 48% for CM and 100% for AN, CA, CV, NA and RM. However, we only were able to compile data on 10 populations for AR (10.5% of the 95 populations surveyed in 2004–2006; Suárez, 2010), the region in which the majority of the Spanish Dupont’s lark population is concentrated (Suárez, 2010). In addition, and regarding the temporal coverage, a high proportion of counts within specific populations are missing. Thus, overall trends (3.9% annual decline rate) may be somewhat biased due to the absence of data throughout the years and for important populations. Therefore, future population trend analyses incorporating a higher proportion of the regional populations in AR are needed, as well as more intensive monitoring to avoid missing temporal data. Accordingly, priority should be given to standardizing and coordinating among populations long-term monitoring, particularly in those large populations in AR.

One additional precaution is related to the lack of fit in models, probably due to missing counts and slight overdispersion in data (i.e. variance greater than the mean). A higher proportion of missing counts leads to greater uncertainty, relying on the statistical model to estimate missing counts. This uncertainty hampers model fitting and may produce population indices that reflect changes in the pattern of missing values rather than real trends (e.g. CA; Pannekoek & Van Strien, 2005). On the other hand, overdispersion could be due to unknown variables not incorporated in the models, which could influence trends (Quinn & Keough, 2002; Crawley, 2007). For instance, interannual variability in population trends encompassed by the significant change-points (Tables S1 and S3) could be explained by natural stochasticity, either demographic or environmental (Lande, 1987), as well as density-dependent interactions (Bjørnstad & Grenfell, 2001). Demographic stochasticity, especially in small and isolated populations, may be an important driver of the observed oscillations between years, since Dupont’s lark seems to fit to a metapopulation structure with local extinction events and colonization processes (e.g. Alfés population in CA; Bota, Giralt & Guixé, 2016). This produces high variability in TRIM yearly indices (i.e. overdispersion), and therefore hinders the estimation of generalized population trends over time. On the other hand, interannual variability may also be associated with environmental stochasticity and fluctuations in abiotic factors, such as climate (Delgado et al., 2009) due to its effects on food availability (Wiens, 1989; Lemoine et al., 2007), reproductive success (Bolger, Patten & Bostock, 2005; Van De Pol et al., 2010) or annual survival (Robinson, Baillie & Crick, 2007), among others. Future research should focus on disentangling the mechanisms underlying variability in trends in order to incorporate new covariates in models and improve their Goodness-of-fit. Regardless, the lack of fit would not invalidate indices, overall slope or Wald tests (Pannekoek & Van Strien, 2005), and consequently the main results regarding Dupont’s lark population trends remain reliable.

We found large differences between regions in population trends; drastic declining trends (annual declining rate higher than 5%) occurred in AN and CL, while trends were classified as uncertain in the other regions (AR, CM, CA, CV, NA and RM). Uncertainty in trends may be due to two typical handicaps in long-term databases: (i) high variability between years and populations (within a region) that produces large CI (i.e. overdispersion); and (ii) high proportion of missing values (Atkinson et al., 2006). As stated above, overdispersion was low except in CA, which could be explained by the extinction–recolonization process undergone by the single population in this region (Bota, Giralt & Guixé, 2016). The percentage of missing values (Table 1) exceeded the recommended threshold of 20–50% for TRIM analyses (Pannekoek & Van Strien, 2005). These two analytical constraints have negligible effects at the national scale but less reliable estimates are expected to be obtained with small-size samples (i.e. regional analysis; Atkinson et al., 2006). The most remarkable case is for the population of CA, where overdispersion, small sample size and a high percentage of missing values prevent the fitting of a switching linear trend model and lead to a mismatch between model-based (i.e. model predictions) and imputed (i.e. observed counts) indices (Fig. 3). Consequently, results for some regional trends should be treated with caution, especially when dealing with a low proportion of populations (i.e. low geographical coverage, e.g. AR; see above) and/or high percentage of missing values (i.e. low temporal coverage).

Inter-region variability in trends may be due to spatial variation in factors threatening Dupont’s lark populations. Declining population trends in AN may be due to agro-forestry (Laiolo & Tella, 2006a) and irrigated land expansion (Íñigo et al., 2008) which have taken place over the last decade. In addition, isolation and small population size make the AN populations more prone to extinction (Méndez, Tella & Godoy, 2011). On the other hand, declining trends for the CL populations can be mainly explained by the implementation of wind farm infrastructures (Gómez-Catasús, Garza & Traba, 2018) or high-speed trains (Íñigo et al., 2008), as well as conifer plantations promoted by the Common Agricultural Policy over marginal low-productivity areas (Tella et al., 2005; Garza & Traba, 2016). The uncertainty in population trends for the other regions makes it difficult to find a potential explanation, although agro-forestry (RM; Laiolo & Tella, 2006a), irrigated lands (AR; Íñigo et al., 2008), afforestations (AR, CV, NA, RM; Tella et al., 2005, Garza & Traba, 2016) and infrastructure development (highways in AR; Íñigo et al., 2008; Garza & Traba, 2016) are among the probable causes. In addition, demographic stochasticity may be a crucial driver of population trend oscillations in small and isolated populations such as AN, CA, NA and RM. In any case, agricultural intensification and abandonment of traditional extensive livestock are general processes known to impact shrub-steppes (Santos & Suárez, 2005), and particularly Dupont’s lark populations (Tella et al., 2005; Íñigo et al., 2008; Garza & Traba, 2016; Gómez-Catasús et al., 2016).

The comprehensive assessment of the conservation status of the Dupont’s lark yielded a higher category of threat according to A2 criterion (future population trends) than A1 criterion (past population trends). The fulfillment of one criterion is enough to classify the species at the highest category of threat. Thus, according to A2 criterion, the Dupont’s lark is correctly listed as ‘Vulnerable’ on the European Red List of Birds, on the SCTS and on the Regional Catalogues of CM, CV and RM. Of particular concern, however, are Dupont’s lark populations in AN and CL, where the species qualifies for listing as ‘Endangered’. However, CL has not yet elaborated a RCTS, while the species is currently listed as ‘Vulnerable’ in AN. In the other regions (AR, CA and NA), the species should be classified as ‘Vulnerable’ according to the category of threat assigned in the SCTS (Law 42/2007, 13th December). If the same assessment had been carried out using previous applicable criteria in the SCTS (before March 2017; Dirección General para la Conservación de la Naturaleza, 2004), the cataloguing scenario would have changed drastically. Under the old criteria, the Dupont’s lark should have been listed as ‘Endangered’ (A2 criterion; population size reduction of ≥40% within the next 20 years), providing evidence of the effects that listing criteria modification may have on the management and conservation of threatened species.

In this study, we assessed the conservation status of the European Dupont’s lark population according to A criteria, since we had no reliable data for including other criteria in our analyses. Therefore, a similar comprehensive assessment should be carried out considering the remaining listing SCTS criteria (reduction in area of occupancy and/or population viability analysis; Resolution 6th March 2017) to elucidate whether or not the species should be classified as ‘Endangered’, ensuring proper listing of the species at both European and national levels. For instance, there is consensus among experts (D criteria; Resolution 6th March 2017) about the need for its reclassification as ‘Endangered’ (Tella et al., 2005; Pérez-Granados & López-Iborra, 2014; Garza & Traba, 2016). Future research should focus on accurately estimating the reduction in area of occupancy. Moreover, a demographic population viability analysis assessing the extinction risk in the coming years should be carried out, although estimating reliable demographic parameters for the whole population of this secretive species is challenging.

Conclusions

Despite methodological constraints due to slight overdispersion, missing data and a low proportion of populations incorporated for AR, we believe that our results in relation to the conservation status of the species in Europe are conclusive. The European Dupont’s lark population faces a 3.9% annual decline rate, entailing an expected average population decline of 32.8% within the next 10 years. The pressures faced by the species have not ceased in recent years (Tella et al., 2005; Íñigo et al., 2008; Garza & Traba, 2016), and may be expected to increase in the future due to strong fragmentation and high vulnerability to stochastic factors (Laiolo & Tella, 2006b; Vögeli et al., 2010; Méndez, Tella & Godoy, 2011). Under this scenario, the implementation of a wide-range conservation plan for the Iberian distribution is vital to ensure the conservation of the species (Íñigo et al., 2008). According to Spanish legislation, the elaboration of a Conservation Plan is mandatory for those species classified as ‘Vulnerable’, such as the Dupont’s lark since 2004 (Orden MAM/2784/2004), and this is within the jurisdiction of the Autonomous Communities. In addition, Autonomous Communities are legally bound to comply with current legislation in cataloguing endangered species (Law 42/2007, 13th December). Therefore, the species should be classified as ‘Endangered’ in AN and CL, and as ‘Vulnerable’ in AR, CA and NA. In this context, the legal responsibility of administrations is crucial to reverse declining population trends of this and other endangered taxa.

Supplemental Information

Supplemental Information 1 Raw data employed in the population trend analysis.

The population size per year (from 2004 to 2015) and per Dupont’s lark population (92 populations) is shown. Number -1 indicates the absence of data for a specific year.

Click here for additional data file.

Supplemental Information 2 Results of the Switching Linear Trend model for 92 Dupont’s lark populations in Spain.

Wald-tests and associated p-values (p) are shown. For each period, the annual change rate, the associated 95% Confidence Interval (CI95%) and trend classification attending to TRIM criteria (TRIM Trend; Pannekoek and Van Strien 2006a) are shown.

Click here for additional data file.

Supplemental Information 3 Local extinction events registered on Dupont’s lark populations from 2004 to 2015 per Autonomous Community.

The number of males at the first time-point (Nt0) and the first year of the temporal series (t0), are shown. In addition, the year when the population was considered extinct or the second year without detecting the species (ti), is indicated.

Click here for additional data file.

Supplemental Information 4 Results of regional Switching Linear Trend models for each Autonomous Community.

Wald-tests and associated p-values (p) are shown. For each period, the annual change rate, the associated 95% Confidence Interval (CI95%) and trend classification attending to TRIM criteria (TRIM Trend; Pannekoek & Van Strien, 2006a) are shown. AN: Andalusia. AR: Aragon. CA: Catalonia. CL: Castile-Leon. CM: Castile-La Mancha. CV: Community of Valencia. NA: Navarre. RM: Region of Murcia.

Click here for additional data file.

The authors wish to thank the administrations of each Autonomous Community and all the people who generously provided data on Dupont’s lark populations in Spain. We are grateful to Ryan Huang and an anonymous reviewer for their constructive comments that helped to improve the manuscript.

Additional Information and Declarations

Competing Interests

Author Contributions

Animal Ethics

Data Availability

The authors declare that they have no competing interests.

Julia Gómez-Catasús performed the experiments, analysed the data, contributed reagents/materials/analysis tools, prepared figures and/or tables, authored or reviewed drafts of the paper, approved the final draft.

Cristian Pérez-Granados performed the experiments, contributed reagents/materials/analysis tools, authored or reviewed drafts of the paper, approved the final draft.

Adrián Barrero performed the experiments, contributed reagents/materials/analysis tools, authored or reviewed drafts of the paper, approved the final draft.

Gerard Bota performed the experiments, contributed reagents/materials/analysis tools, authored or reviewed drafts of the paper, approved the final draft.

David Giralt performed the experiments, contributed reagents/materials/analysis tools, authored or reviewed drafts of the paper, approved the final draft.

Germán M. López-Iborra performed the experiments, contributed reagents/materials/analysis tools, authored or reviewed drafts of the paper, approved the final draft.

David Serrano performed the experiments, contributed reagents/materials/analysis tools, authored or reviewed drafts of the paper, approved the final draft.

Juan Traba conceived the experiments performed the experiments, contributed reagents/materials/analysis tools, authored or reviewed drafts of the paper, approved the final draft.

The following information was supplied relating to ethical approvals (i.e. approving body and any reference numbers):

The ethics committee of Animal Experimentation of the Autonomous University of Madrid as an Organ Enabled by the Community of Madrid (Resolution 24th September 2013) for the evaluation of projects based on the provisions of Royal Decree 53/2013, 1st February, has provided full approval for this purely observational research (CEI 80-1468-A229).

The following information was supplied regarding data availability:

The raw data are provided in the Supplemental File.

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
