# Peer review of "European population trends and current conservation status of an endangered steppe-bird species: the Dupont’s lark Chersophilus duponti"

_PeerJ, doi:10.7717/peerj.5627_

## Round 0.1 · original submission · Major Revisions

· Academic Editor

Major Revisions

I agree with both the reviewers, who incidentally, raise much the same concerns. To resolve these issues will require major revisions.

I look forward to your revised manuscript

·

Basic reporting

The writing style is professional and the grammar is correct with the exception of the incorrect and overuse of the word “besides” to start sentences.

The figures and tables are appropriate and clean, however I would suggest combining S3-S10 into a single table with an additional column for the Autonomous Region.

Experimental design

The authors have clearly established the need for an updated assessment of the Dupont’s lark at both the regional and national level. While a large number of population assessments have been used, the author’s don’t seem to use all of them and it’s not clear why not.

The extinction event information is very interesting and it’s good to include. That being said, there should be a sentence defining local extinction events in the methods and why you don’t simply assume it’s due to poor detection rather than true local extinction (i.e. it requires multiple years of no detection to be considered a local extinction).

Validity of the findings

The raw data is clean, but is clearly missing many years as noted by the authors. Given the missing data and the variety of sites, TRIM appears to be a reasonable software to use to analyze the data. That being said, I’m not sure I believe the results for Catalonia. There is a single population underwent an extinction and recolonization event, but Nti is smaller than Nt0, and yet the results indicate a positive growth trend? Despite being classified as an “uncertain” trend and the author’s note that overdispersion is a concern, this doesn’t make intuitive sense. Why isn’t there at least one change point in the middle to represent recolonization? Unfortunately, because I doubt one of these models, I now doubt the remainder.

The discussion covers much of the necessary topics but could be expanded some. For instance, demographic stochasticity is mentioned, but I don’t think is given the appropriate weight. Habitat loss may reduce population size through the reduction of resources, but in the absence of total patch removal, demographic stochasticity is likely the driver of local extinction of these small populations. Additionally, the authors discuss interannual variability but don’t have any mention of inter-site variability. Why do some Autonomous Regions see more decline than others? Is it solely due to rates of habitat loss? This is an important topic to discuss in light of the results and differential listings for the Dupont’s lark.

Additional comments

• The authors have clearly stated the need for an updated assessment of the Dupont’s lark and appear to have done the most comprehensive analysis to date. That being said, it is not clear why despite mentioning the number of population counts performed, they couldn’t use more data. This can be easily clarified with a single sentence in the methods.

• The larger concern is the use of a “black box software” for estimating population declines. While TRIM appears to be a valid and appropriate analytical software given the variety of sites and missing data, I’m not convinced the outputs make intuitive sense given the raw data. This is the problem with these types of software, one can simply input all of their data and a result is spit out and no one can be entirely sure where this number came from. My recommendation would be a secondary set of analyses using basic population viability analyses to compare to the TRIM results.

• Lastly, the Discussion could be expanded some more. In particular, expounding on potential site differences and what’s driving differences in vulnerability and population declines.

Reviewer 2 ·

Basic reporting

see attached pdf

Experimental design

see attached pdf

Validity of the findings

see attached pdf

Additional comments

see attached pdf

Annotated reviews are not available for download in order to protect the identity of reviewers who chose to remain anonymous.

---

## Round 0.2 · accepted · Accept

· Academic Editor

Accept

Under normal circumstances, with a previous decision of "major revisions," I would send this back to the reviewers for another round of review. In this case, I know that both my reviewers are travelling and do not have easy internet access. I'm afraid that if I send it out again it might be some considerable time before I get comments back.

Secondly, I had the chance to discuss the reviews and I shared their principal concerns about inferences about trends when data are missing.

Having now read your paper carefully and noted the very thorough point-by-point response to the earlier concerns, I have decided to accept this revision without further review. I think you've done an excellent job at addressing the statistical concerns. The paper as a whole makes a compelling case that this species is in serious trouble and that immediate actions are needed to conserve it.

My only request is an entirely personal one! Given how secretive this species is even when singing, is there any chance at all in seeing it in November? (That's the only time I'm going to be near Spain in the near future.)

#

---

## Author Rebuttal · Round 0.2

**Dear Dr. Stuart Pimm**,

Thank you for your invitation to prepare a revision of the first version of our manuscript. We have considered all your suggestions and the constructive comments of both reviewers. We have detailed all the changes we have made (Manuscript_text_changes_eng.doc) and we have answered the issues highlighted by each reviewer. First, we answer in detail to the main topics and suggestions, to then address point by point the reviewers' concerns.

Both reviewers were concerned about the **number of populations** included in the population trend analysis. Our dataset consists of 92 Dupont's lark populations, which comprises 41.6% of the whole Spanish population known during the II (and most recent) National Survey by Suárez (2010). This reference is included in the manuscript to support the representativeness of our dataset, but no direct data was taken from this study. Data from the 92 populations included in our analyses were directly sampled by the authors or collaborators. The remaining Iberian populations surveyed by Suárez (2010) were not included in our study because we had no monitoring data over time. In the revised version of the manuscript, this section has been rephrased (see Methods) and we hope it is now much more clear. We consider that our dataset of 92 populations is good enough to obtain conclusive trends of Dupont's lark in Spain.

Secondly, both reviewers were concerned about the **proportion of gaps in the data**. In accordance, we have expanded the discussion about the effect of missing counts on population trend results. In addition, as Reviewer 2 suggested, we have repeated the analysis using only those populations with data for at least half of the number of years considered in the study (i.e. 6 years). New results suggest a 4.97% average annual decline for the whole Spanish (European) population, 1.27 times more regressive than our initial result (3.9% average annual decline with the whole dataset). At the regional scale, population trends remained similar to previous results. However, regional population trends for Catalonia (CA) and Navarre (NA) could not be assessed due to the absence of populations meeting the 6-years criteria. In the light of these results, we consider that original results are more conservative and representative of the whole Spanish population (92 populations against 52 populations in the new analysis), covering all regions where the species is currently present. Therefore, we have preferred to keep the analyses with all 92 populations in the

revised version of the manuscript and have not included the new ones to do not enlarge the manuscript excessively. However, we have attached a document incorporating the new results (Review-only_info.doc) for further evaluation by reviewers and editor, and we are willing to incorporate it as Supplementary Material if you consider it appropriate.

On the other hand, Reviewer 1 was concerned about the **results obtained for Catalonia region**. In this sense, we apologize since previous results were improvable. When dealing with high proportion of missing counts and small sample size (i.e. small number of populations), TRIM designers recommend employing imputed and not model indices. Imputed indices equal the observed count if a real observation was available and the predicted values from the model when real observations are missing. Instead, model-based indices are all model predictions irrespective of whether or not real counts are available. These two indices and associated standard errors hardly differ in most regions. However, they differ widely in Catalonia due to the high proportion of missing values, small sample size (just one population) and high interannual variability in trends (extinction and recolonization events). In the previous manuscript, we employed the model-based indices, obtaining an average annual increase of 3.6% in CA. In this new manuscript, we have reanalyzed the whole dataset and depicted the imputed indices, obtaining a 8.7% average annual decline in CA. We consider that this result is more realistic and reliable, and it addresses the reviewer's concern for the mismatch between the (previous) positive trend inferred and the observed decline (from 7 males in $N_0$ to 4 in $N_t$). We have incorporated the definition of imputed and model-based indices in Methods, and we have depicted both indices in Figures 1 and 2 as a visual representation of model fit. Lastly, we have depicted the 95% Confidence Intervals in order to represent the uncertainty associated to the estimates.

Still on the subject of **Catalonia results**, the reviewer wondered why a change-point was not incorporated into the model to represent the recolonization event. This is a statistical drawback because when data are missing or sparse, a Switching Linear Trend model cannot be fitted and a Linear Trend Model must be fitted instead. To solve this, we have incorporated a few lines in Results about the absence of change-points in Catalonia due to scarce data, which prevented us from fitting a Switching Linear Trend model.

Lastly, Reviewer 1 raised his concern about using a **"black box software" for estimating declining population trends**. TRIM has been widely employed for the analysis of temporal series

in bird populations (e.g. Paradis *et al.* 2002, Wretenberg *et al.* 2007, Delgado *et al.* 2009, Gómez-Catasús et al. 2018). This software has two advantages above other statistical tools. First, it allows to analyse time series with absence of data in some years, a common issue in long-time series. Secondly, it takes into account overdispersion and serial correlation of data. It is true than in the presence of small sample sizes, high overdispersion and high percentage of missing values (e.g. Catalonia model), the statistical power decreases and results may not be fully reliable. But neither they are when fitting, for example, a classic log-linear model. Therefore, we consider that TRIM is an appropriate statistical tool to obtain population trends and that results are reliable in most of our analysis. When results were not fully reliable (e.g. Catalonia model), these handicaps (small size, overdispersion and missing data) have been discussed. Regarding the recommendation of using a population viability analysis, we use a quantitative methodology to predict the likely future status of a collection of populations of conservation concern, which is precisely the broad definition of a PVA (see Morris F & Doak DF. 2002. Quantitative Conservation Biology. Theory and practice of population viability analysis. Sinauer Associates Inc., Sunderland, Massachusetts). Indeed, Morris and Doak dedicated nothing less than three full chapters to count-based PVAs in their seminal book. We are well aware that this kind of approach has important limitations, but more sophisticated PVAs, such as matrix models fed with demographic data, are not problem-free either. In the case of the Dupont's lark, its secretive and elusive behavior, the spatial structure of their populations and the tiny size of many of them make it difficult to collect vital rates and their spatiotemporal variability. On the other hand, maybe the Reviewer refers to the common practice in PVAs of providing any measure of risk assessment (e.g. extinction risk) and/or of predicting population trajectories under different management scenarios. These practices are however out of the scope of this manuscript, which tries to draw attention to the actual negative population trends of the species and if these trends match the listing categories.

Other aspects that were missing and highlighted by the reviewers have been incorporated in this new version of the manuscript: more details about the survey methods, discussion about the effect of missing counts, discussion about interpopulation variability in population trends and demographic stochasticity, among others. These changes can be followed in the tracked changes manuscript (Manuscript_text_changes_eng.doc).

Finally, the manuscript has been checked by a native English in order to solve the errors in English usage and grammar. We hope that the manuscript is now easier to follow.

We think that this new manuscript improves the previous one and clarifies the weakest points. We answer to each reviewer below on a point-by-point basis.

Looking forward to hearing from you and thanking you for your work as editor.

Sincerely,

Julia Gómez-Catasús and co-authors

**Dear Dr. Ryan Huang**,

Thank you for your kind comments and suggestions. We have paid attention to the specific and general comments that you made and they have really helped us to improve the previous manuscript. We will address first the general and more relevant comments that you made, and then we answer point-by-point your specific suggestions.

1) **Number of populations employed.** We have improve the description about the number of populations included in the population trend analysis. Our dataset consists of 92 Dupont's lark populations, which comprises 41.6% of the whole Spanish population known during the II (and most recent) National Survey by Suárez (2010). This reference is included in the manuscript to support the representativeness of our dataset, but no direct data was taken from this study. Data from the 92 populations included in our analyses were directly sampled by the authors or collaborators. The remaining Iberian populations surveyed by Suárez (2010) were not included in our study because we had no monitoring data over time. In the revised version of the manuscript, this section has been rephrased (see Methods) and we hope it is now much more clear. We consider that our dataset of 92 populations is good enough to obtain conclusive trends of Dupont's lark in Spain.

2) **Robustness of the results obtained for Catalonia region**. In this sense, we apologize since previous results were improvable. When dealing with high proportion of missing counts and small sample size (i.e. small number of populations), TRIM designers recommend employing imputed and not model indices. Imputed indices equal the observed count if a real observation is available and the predicted values from the model when real observations are missing. Instead, model-based indices are all model predictions irrespective of whether or not real counts are available. These two indices and associated standard errors hardly differ in most regions. However, they differ widely in Catalonia due to the high proportion of missing values, small sample size (just one population) and high interannual variability in trends (extinction and recolonization events). In the previous manuscript, we employed the model-based indices, obtaining an average annual increase of 3.6% in CA. In this new manuscript, we have reanalyzed the whole dataset and depicted the imputed indices, obtaining a 8.7% average annual decline in CA. We consider that this

result is more realistic and reliable, and it addresses the you concern for the mismatch between the (previous) positive trend inferred and the observed decline (from 7 males in $N_0$ to 4 in $N_t$). We have incorporated the definition of imputed and model-based indices in Methods, and we have depicted both indices in Figures 1 and 2 as a visual representation of model fit. Lastly, we have depicted the 95% Confidence Intervals in order to represent the uncertainty associated to the estimates.

Still on the subject of **Catalonia results**, you wondered why a change-point was not incorporated into the model to represent the recolonization event. This is a statistical drawback because when data are missing or sparse, a Switching Linear Trend model cannot be fitted and a Linear Trend Model must be fitted instead. To solve this, we have incorporated a few lines in Results about the absence of change-points in Catalonia due to scarce data, which prevented us from fitting a Switching Linear Trend model.

3) **The employment of a "black box software" for estimating declining population trends**. TRIM has been widely employed for the analysis of temporal series in bird populations (e.g. Paradis *et al.* 2002, Wretenberg *et al.* 2007, Delgado *et al.* 2009, Gómez-Catasús et al. 2018). This software has two advantages above other statistical tools. First, it allows to analyse time series with absence of data in some years, a common issue in long-time series. Secondly, it takes into account overdispersion and serial correlation of data. It is true than in the presence of small sample sizes, high overdispersion and high percentage of missing values (e.g. Catalonia model), the statistical power decreases and results may not be fully reliable. But neither they are when fitting, for example, a classic log-linear model. Therefore, we consider that TRIM is an appropriate statistical tool to obtain population trends and that results are reliable in most of our analysis. When results were not fully reliable (e.g. Catalonia model), these handicaps (small size, overdispersion and missing data) have been discussed. Regarding the recommendation of using a population viability analysis, we use a quantitative methodology to predict the likely future status of a collection of populations of conservation concern, which is precisely the broad definition of a PVA (see Morris F & Doak DF. 2002. Quantitative Conservation Biology. Theory and practice of population viability analysis. Sinauer Associates Inc., Sunderland, Massachusetts). Indeed, Morris and Doak dedicated nothing less than three full chapters to count-based PVAs in their seminal book. We are well aware that this kind of approach has

important limitations, but more sophisticated PVAs, such as matrix models fed with demographic data, are not problem-free either. In the case of the Dupont's lark, its secretive and elusive behavior, the spatial structure of their populations and the tiny size of many of them make it difficult to collect vital rates and their spatiotemporal variability. On the other hand, maybe the Reviewer refers to the common practice in PVAs of providing any measure of risk assessment (e.g. extinction risk) and/or of predicting population trajectories under different management scenarios. These practices are however out of the scope of this manuscript, which tries to draw attention to the actual negative population trends of the species and if these trends match the listing categories.

We hope that this new manuscript improves the previous one and that clarifies the weakest points. Looking forward to hearing from and thanking you for your work as reviewer.

Sincerely,

Julia Gómez-Catasús and co-authors
* * *
**POINT – BY – POINT specific comments.** Line numbers refers to the tracked changes masnucript (Manuscript_text_changes_eng.doc).

**L88 – L90** Be careful with the term "umbrella species." How much does the Dupont's lark range overlap with other steppe endemics? Is there evidence or a publication for this?

➢ **L94 – 96:** We have removed this sentence since, as the reviewer suggested, the Dupont's lark is not a strictly umbrella species.

**L119** This is redundant, you have already defined Autonomous Communities as the regional scale

➢ **L127 – 128:** Removed

**L133** How are habitat patches defined?

We rephrased the sentence to define habitat patch as:

➢ **L144 – 147:** "*We considered a single population to be all individuals living in patches with potential habitat for the species (i.e. short shrub with slopes lower than 15%; Garza et al. 2005) separated by less than 1 km.*"

**L134-135** 12 years is a rather long time interval to get any conclusive information on population trends

➢ We agree with the reviewer, as we think that this time interval allows us to get rather conclusive results about the trend of the species.

**L136** Besides
- ➢ **L150:** Removed

**L208** Is an extinction event any time a population count reached 0?

In the previous manuscript, we considered an extinction event any time a population count reached 0. We have now described a local extinction event as two surveyed years without detection. We have incorporated the sentence below in methods to clarify this:

- ➢ **L172 – 174**: "*Lastly, we considered a population as extinct when the species was not detected in at least the last two surveys (hereafter local extinction event).*"

Accordingly, the number of local extinction events at the national and regional levels have been modified in results and tables.

**L208** Besides
- ➢ **L241:** Removed, now it reads "*Furthermore*"

**L223** I would consider combining S3-S10 into a single supplemental table for ease of reading.

As the reviewer suggested, we have combined S3-S10 into a single supplemental table named Table S3.

**L283** You might also want to mention that small sample sizes/populations are more impacted by demographic stochastic effects.

We have incorporated the sentence below to highlight the impact of demographic stochasticity on small populations:

- ➢ **L340:** "*[…] especially in small and isolated populations […]*".

**L300** Besides
- ➢ **L362:** Removed

**REVIEWER 2**

**Dear Reviewer**,

We appreciate your comments. At the end of this letter, you will find point-by-point the answer to your suggestions. However, we will answer here to the main concerns about the number of populations employed and the robustness of the results:

1) **Number of populations employed.** We have improve the description about the number of populations included in the population trend analysis. Our dataset consists of 92 Dupont's lark populations, which comprises 41.6% of the whole Spanish population known during the II (and most recent) National Survey by Suárez (2010). This reference is included in the manuscript to support the representativeness of our dataset, but no direct data was taken from this study. Data from the 92 populations included in our analyses were directly sampled by the authors or collaborators. The remaining Iberian populations surveyed by Suárez (2010) were not included in our study because we had no monitoring data over time. In the revised version of the manuscript, this section has been rephrased (see Methods) and we hope it is now much more clear. We consider that our dataset of 92 populations is good enough to obtain conclusive trends of Dupont's lark in Spain.

2) **Robustness of the results**. We have repeated the analysis using only those populations with data for at least half of the number of years considered in the study (i.e. 6 years). New results suggest a 4.97% average annual decline for the whole Spanish (European) population, 1.27 times more regressive than our initial result (3.9% average annual decline with the whole dataset). At the regional scale, population trends remained similar to previous results. However, regional population trends for Catalonia (CA) and Navarre (NA) could not be assessed due to the absence of populations meeting the 6-years criteria. In the light of these results, we consider that original results are more conservative and representative of the whole Spanish population (92 populations against 52 populations in the new analysis), covering all regions where the species is currently present. Therefore, we have preferred to keep the analyses with all 92 populations in the revised version of the manuscript and have not included the new ones to do not enlarge the manuscript excessively. However, we have attached a document incorporating the new results

(Review-only_info.doc) for further evaluation by reviewers and editor, and we are willing to incorporate it as Supplementary Material if you consider it appropriate.

Finally, the manuscript has been checked by a native English in order to solve the errors in English usage and grammar. We hope that the manuscript is now easier to follow.

We hope that this new manuscript improves the previous one and that clarifies the weakest points. Thank you for your work.

Sincerely,

Julia Gómez-Catasús and co-authors
* * *
**POINT – BY – POINT specific comments.** Line numbers refers to the tracked changes manuscript (Manuscript_text_changes_eng.doc).

**Abstract**

**L23.** End sentence and start a new one after 'unknown'.
We have end the sentence after unkown.

➤ **L24 – 25:** Removed: "*so an updated and rigorous assessment is needed.*"

**L26.** 'to date'
➤ **L27:** Done, now it reads "*to date*"

**L34.** You need to explain, even if briefly, what is lambda, for readers not familiar with TRIM or analyses of this kind.

We consider that this information is too specific to be incorporated in the abstract. However, we have incorporated a description of lambda in Methods:

➤ **Materials and Methods. L221 – 225:** "*The average finite annual rate of change ($\bar{\lambda}$) during the study period was obtained from the TRIM analysis. This is a multiplicative factor representing the average growth rate over one time-step (i.e., one year). When this multiplicative factor is $\bar{\lambda} < 1$ the population decreases; when $\bar{\lambda} = 1$ the population remains stable; and when $\bar{\lambda} > 1$ the population increases.*"

**L40-44** can be removed from the abstract. Not relevant here.
➤ **L43 – 46:** Removed

**L45.** According, not attending.
➤ **L47:** Done, now it reads '*According*'

**L51**. End sentence and start a new one removing 'besides'.
➤ **L55:** Removed

**L52.** 'administrations to law enforcement' doesn't make much sense, please reword.

> **L55 – 56:** Rephrased as "*The role of administrations in matters of nature protection and […]*"

## Introduction

**L57.** 'bird diversity,…'

> **L61:** A comma was incorporated after '*bird diversity*'

**L91.** Assessed instead of addressed

> **L97:** Done, now it reads '*assessed*'

**L93.** Showing

> **L99:** Now it reads '*showed*'

**L106.** In addition to this… not 'besides'

> **L113 - 114:** Done, now it reads '*In addition to this*'

**L113.** Accommodate those of the IUCN

> **L121:** Done, now it reads '*to accommodate those of the IUCN*'

**L122.** Remove 'current applicable criteria in'

> **L130 – 131:** Removed '*current applicable criteria in*'

## Materials and methods

**L128-131.** can be removed.

> **L136 – 139:** We decided to keep this sentence since when submitting the review manuscript we have to confirm that the name of the approval organization and approval number appear in Methods.

**L132-133.** 'We considered a single population to be all individuals living in habitat patches…'

> **L144-145:** Replaced '*We considered a population as those habitat patches*' by '*We considered a single population to be all individuals living in patches*'

**L134.** Accounted for -> comprised

> **L141:** Done, now it reads '*comprised*'

**L136.** What do you mean by 'besides'? Are there any other populations considered? Clarify.

> **L150 - 151:** We have removed this sentence in order to avoid confusion. We mean that the regional analysis was carried out for all the Autonomous Communities where the species is present (there are more communities but the regional analysis have not been carried out for those because the species does not occur).

> **L142 – 143**: Instead, we have clarified it writing '[…] *and included all of the Autonomous communities where the species occurs* […]'.

**L140-151.** For the bird censuses you need to specify how did you do the counting: everyday? For how long?

This depends on the census method employed. Under the linear transect method the counting was done just once per year and under the mapping method it was done between 2 and 4 times per year.

> ➤ **L166:** We have incorporated the words "*per year*" after "*once*" and "*2-4 times*" to clarify.
> ➤ **L156 – 157:** We have incorporated information about the duration of surveys: "*and they spanned around 1 hour.*"

Also, it is not clear if you did this counting for all the 92 populations or some data were used as they were from other studies. This must be clarified. Specify exactly how many populations you sampled.

As we stated above, we carried out the counting for all the 92 populations. No information was used from the study of Suárez (2010). With this reference we wanted to support the representativeness of our dataset, which comprises 41.6% of the whole Spanish population known during the II (and most recent) National Survey by Suárez (N=221 populations). The remaining populations in Suárez (2010) were not incorporated in this study because we had no monitoring data over time. This section has been rephrased in methods for a better understanding and to avoid confusion:

> ➤ **L140 – 144:** "*We compiled data for 92 Dupont's lark populations during the 2004-2015 period. This dataset comprised 41.6% of the known Spanish population (221 populations surveyed during the II National Survey 2004-2006; Suárez 2010) and included all of the Autonomous communities where the species occurs (Fig. 1) (Suárez 2010). The time series addresses a temporal range between one and 12 years (mean ± SD = 5.36 ± 2.77 years).*"

What did you do with the repeated observations? Sum? Max? avg?

With the repeated observations (in the case of the mapping method) we used the territory mapping method to locate male territories and estimate population sizes. This method provides accurate results when studying territorial passerine species (Bibby *et al.* 2000). Territories were delimited by gathering accumulated observations from different surveys and interpreting simultaneously contacted neighbouring males. Despite the effort, this method provides similar results than carrying out only one census in the right moment of the season (Pérez-Granados and López-Iborra 2017).

> ➤ **L167 – 172:** We rephrased this sentence as:
>
> "*In the case of the territory mapping method, number of territories per population was estimated by mapping all records and gathering accumulated observations from different surveys, taking into account birds heard simultaneously (Garza et al. 2010, Pérez-Granados and López-Iborra 2017). Population size estimates refer to the minimum number of territories (mapping method), or minimum number of recorded males (line transect method) per population*"

**L145.** Consisting of

➢ **L160:** Done, now it reads '*consisting of*'

**L147.** Besides -> in any case
➢ **L162:** Removed '*Anyway*'

**L148.** Designed not designated.
➢ **L163:** Done, now it reads '*designed*'

**L152.** Records not registrations.
➢ **L169:** Removed '*registrations*'

**L172.** What do you mean by 'attending'? complying with?
➢ **L200:** Done, now it reads '*complying with*'

**L191.** Annual rate of what?
➢ **L221:** Clarified, now it reads '*The average finite annual rate of change*'

**L194.** Why do you subtract 1 from the rate powered to the number of years in this equation?

Lambda is not the rate of change, is the finite rate of change. This means that lambda is a multiplicative factor or, in other words, the average per-capita multiplication factor per one time-step (one year in our study). When $\lambda < 1$ the population decreases; when $\lambda = 1$ the population remains stable and when $\lambda > 1$ the population increases. We have to subtract 1 from the finite rate of change to obtain the percentage of change. For instance, when $\lambda = 0.80$ the population undergoes a (0.80 - 1) * 100 = -20% average annual decline. If lambda is raised to the powered of 10, we estimate the finite rate of change in a 10-year period; $\lambda^{10} = 0.80^{10} = 0.107$. This means that the population undergoes a (0.107 – 1 ) * 100 = - 89% decline in 10 years.

➢ **L221 – 225:** We have incorporated a brief defnition of lambda to clarify this misunderstanding:

"*The average finite annual rate of change ($\bar{\lambda}$) during the study period was obtained from the TRIM analysis. This is a multiplicative factor representing the average growth rate over one time-step (i.e., one year). When this multiplicative factor is $\bar{\lambda} < 1$ the population decreases; when $\bar{\lambda} = 1$ the population remains stable; and when $\bar{\lambda} > 1$ the population increases*"

**Results**
**L208.** Besides -> furthermore
➢ **L241:** Done, now it reads '*Furthermore*'

**L228-229.** Table 2 not 1.
➢ **L264 - 265:** This references are to table 1, showing the results of regional Switching Linear Trend models. Table 2 refers to the assessment of Dupont's lark threat category.

**L229.** Delete 'Fig. 3' as this statement doesn't have to do with that figure.
➢ **L266:** Removed

**L230-232.** Hard to understand what you mean. Please rephrase.

> **L269 – 271:** To avoid confusion, we have removed the sentence "*Frequency of local extinction events were higher in CA (100%, only one population under study which was ultimately recolonized in 2015), AN (50%), CV (37.5%), NA (33.3%), CL (31%) and CM (23.1%)*". The frequency of local extinction events is dependent on the number of studied populations and not on the total number of populations. Therefore, it says nothing about the real frequency of extinction in each region and it only leads to confusion.

**L231-232.** The statement between brackets is misleading because you didn't visit this site between 2008-2014 so you can't know if this population was actually extinct or if it came back before 2015. Please remove. Also, attributing a population extinction rate of 100% for this region based on a single population that was not actually visited during most of the period is misleading. You need to state clearly the circumstances of this or remove this statement.

> **L269 – 270:** Removed

**L236.** Will be -> we expect it to be
> **L274-275:** Done, now it reads '*we expect it to be*'

**L241.** 'classify' -> 'be classified'
> **L279:** Done, now it reads '*be classifed*'

**L242.** 'and' -> ',while'
> **L280:** Done, now it reads '*while*'

**Discussion**

**L250-253.** What are these trends? Mention them here and expand this discussion.

We have expand the discussion supporting our results in previous works about Dupont's lark population trends:

> **L287 – 303:** We have incorporated:

'*The species exhibited an estimated annual decline rate of 3.9% over the last decade and a 41.4% decline over 12 years (2004 – 2015).This result agrees with previously described trends for the Dupont's lark (Tella et al. 2005, Pérez-Granados and López-Iborra 2013) and for most of steppe-bird species in the Iberian Peninsula (Burfield 2005, BirdLife International 2015).Previous work on Spanish Dupont's lark population trends suggested a 31.5% decline in 16 years (N = 34 populations; Tella et al. 2005) and a 70% decline in 12.5 years (N = 33 populations; Pérez-Granados and López-Iborra 2014). In particular areas of its Spanish distribution, positive trends have been previously estimated in Aragon (N=7) and Region of Murcia (N=2), whereas declining population trends have been described for Andalusia (N=4), Castile-La Mancha (N=6), Castile-Leon (N=6), Community of Valencia (N=6) and Navarre (N=2) populations (a decline between 22% and 98% decline in 12.5 years; Pérez-Granados et al. 2014). The novelty of the present work relies on the employment of a rigorous statistical method and on the incorporation of a greater number of Dupont's lark populations (N = 92) covering a wider range of its European distribution.*'

**L254-261.** This statement doesn't seem to make much sense here. It is isolated from

the flow of the discussion and doesn't make any point to the argument being developed.

➢ **L303 – 312:** Removed

**L262.** I wouldn't call this database exhaustive.

➢ **L313:** Removed '*exhaustive*'.

**L265-266.** This statement is misleading due to missing temporal data. You need to provide fractions of actual sampling events. For example, I wouldn't catalogue CA as being 100% represented in the sample if it wasn't visited during 7 out of the 12 years of the study period.

As the reviewer suggested, our dataset lack of temporal representation due to the high percentage of missing data within populations. We kept the information about the geographical coverage because we consider that this information is also relevant. In addition, we have incorporated three sentences to highlight the weakness of our dataset regarding the temporal coverage (for specific populations):

➢ **L320 – 322:** '*In addition, and regarding the temporal coverage, a high proportion of counts within specific populations are missing.*'
➢ **L322 – 324:** '*Thus, overall trends (3.9% annual decline rate) may be somewhat biased due to the absence of data throughout the years and for important populations.*'
➢ **L325 – 326:** "*as well as more intensive monitoring in order to avoid missing temporal data.*"

**L274-282.** I think a major factor influencing this is the gaps in the data. You should discuss it here along with the more appealing biological and ecological factors.
We have expanded the discussion about the effect of gaps in the data:

➢ **L329 – 336:** "*One additional precaution is related to the lack of fit in models, probably due to missing counts and slight overdispersion in data (i.e. variance greater than the mean). A higher proportion of missing counts leads to greater uncertainty, relying on the statistical model to estimate missing counts. This uncertainty hampers model fitting and may produce population indices that reflect changes in the pattern of missing values rather than real trends (e.g. CA; Pannekoek and Van Strien 2005). On the other hand, overdispersion could be due to unknown variables not incorporated in the models, which could influence trends (Quinn and Keough 2002, Crawley 2007)*"